# Cardiovascular Effect of Epoprostenol and Intravenous Cardiac Drugs for Acute Heart Failure on Canine Pulmonary Hypertension

**DOI:** 10.3390/vetsci10040302

**Published:** 2023-04-19

**Authors:** Yunosuke Yuchi, Ryohei Suzuki, Shuji Satomi, Takahiro Saito, Takahiro Teshima, Hirotaka Matsumoto

**Affiliations:** 1Laboratory of Veterinary Internal Medicine, School of Veterinary Science, Faculty of Veterinary Medicine, Nippon Veterinary and Life Science University, Tokyo 180-8602, Japan; y.0301.yunosuke@gmail.com (Y.Y.); jetlog21117@yahoo.co.jp (S.S.); justthe2ofussaito@yahoo.co.jp (T.S.); teshima63@nvlu.ac.jp (T.T.); matsumoto@nvlu.ac.jp (H.M.); 2Garden Veterinary Hospital, Tokyo 153-0063, Japan

**Keywords:** cardiac output, dog, prostaglandin I_2_, pulmonary arterial pressure, pulmonary arterial resistance, right heart catheterization, right heart failure, speckle tracking echocardiography

## Abstract

**Simple Summary:**

Pulmonary hypertension is a life-threatening complication in dogs with cardiopulmonary disease. Epoprostenol, a prostacyclin analog, is an intravenous pulmonary vasodilator used for treating pulmonary hypertension in humans; however, it has not been used in veterinary medicine. This study aimed to investigate the effects of epoprostenol on cardiac function and hemodynamics in canine models of chronic pulmonary hypertension. Six laboratory-owned dogs with chronic embolic pulmonary hypertension were anesthetized and underwent right heart catheterization and echocardiography before and after administering epoprostenol, dobutamine, dopamine, and pimobendan. In this study, high doses of epoprostenol significantly decreased pulmonary and systemic vascular resistance and increased left and right ventricular function. In contrast, dobutamine and dopamine significantly increased pulmonary arterial pressure and left and right ventricular function. Pimobendan significantly increased left and right ventricular function without increasing pulmonary arterial pressure. Our results indicate that high doses of epoprostenol might help effectively treat canine pulmonary hypertension through pulmonic and systemic vasodilating effects. Catecholamines improve left and right ventricular function; however, they might worsen the pathophysiology of pulmonary hypertension. Pimobendan also improved left and right ventricular function without worsening pulmonary hypertension pathophysiology; however, a stronger vasodilating effect was observed with epoprostenol.

**Abstract:**

Pulmonary hypertension (PH) is a life-threatening complication in dogs with cardiopulmonary disease. Epoprostenol is an intravenous pulmonary vasodilator used to treat PH in humans; however, its efficacy in dogs remains unknown. We investigated the cardiovascular effects of epoprostenol and several cardiac agents for acute heart failure in canine models of chronic PH. Six dogs with chronic PH were anesthetized and underwent right heart catheterization and echocardiography before and after infusion of epoprostenol, dobutamine, dopamine and pimobendane. (The drug administration order was the same for all dogs). High-dose epoprostenol (15–20 ng/kg/min) tended to decrease pulmonary arterial pressure (PAP) while significantly decreasing pulmonary and systemic vascular resistance and increasing left and right ventricular (LV and RV, respectively) function. Pimobendan significantly increased LV and RV functions without increasing PAP. Conversely, dobutamine and dopamine significantly increased LV and RV function as well as PAP. This study revealed the efficacy of epoprostenol in treating canine PH through its pulmonary and systemic vasodilating effects. Although catecholamines improve LV and RV function, they might worsen PH pathophysiology, and careful monitoring may be necessary when using these drugs. Pimobendan improved LV and RV function without increasing PAP; however, a stronger vasodilating effect was observed with epoprostenol.

## 1. Introduction

Pulmonary hypertension (PH) is a life-threatening disease in dogs that is characterized by an increase in pulmonary arterial pressure (PAP) and/or pulmonary vascular resistance (PVR) [1,2]. This disease may induce right ventricular (RV) hypertrophy, dysfunction and dilatation through excessive RV afterloads. Finally, dogs with PH may develop lethal clinical signs, such as syncope, ascites, pleural effusion and multiple organ failure due to circulation insufficiency and right heart congestion. In dogs with cardiopulmonary disease, concomitant PH has been reported to be associated with poor outcomes [1,3,4]. Therefore, optimal diagnosis, pathophysiological evaluation and therapeutic monitoring of PH have received increasing attention in veterinary medicine.

The American College of Veterinary Internal Medicine has recently established guidelines for diagnosing and treating PH in dogs [1], which has helped to unify the perspectives on canine PH in veterinary medicine. This guideline divides canine PH management into three strategies: reducing the triggers and exacerbators of PH, treating the underlying diseases attributable to PH and PH-specific treatments. However, no information on how to deal with acute exacerbations of PH is described in the guidelines. Currently, several cardiac drugs, such as catecholamines (e.g., dobutamine and dopamine) and phosphodiesterase-3 inhibitors (e.g., pimobendan), are used to treat acute heart failure associated with PH in veterinary clinical practice. Conversely, since the effectiveness of catecholamines in treating PH has not been reported in veterinary medicine, they have been used empirically in the same way as in dogs with acute left-sided heart failure. Regarding pimobendan, recent studies have reported the effectiveness of intravenous and intramuscular pimobendan for the treatment of PH in dogs [5,6]. However, because most dogs already use oral pimobendan to treat PH in the chronic phase, it remains unclear whether intravenous pimobendan has any additional benefit during acute exacerbations of chronic PH.

In human medicine, epoprostenol, an intravenous prostaglandin I_2_ (prostacyclin) analog, has been used to treat PH and is the only intravenous agent highly recommended for patients with severe PH [7,8]. The prostaglandin I_2_ analog has various effects on pulmonic and systemic hemodynamics, including the protective effect on endothelial cells, as well as pulmonary and systemic vasodilating, anti-inflammatory and anti-platelet effects [9,10,11,12,13]. In veterinary medicine, we previously reported the efficacy of an oral prostaglandin I_2_ analog (beraprost sodium) in canine PH [14,15]. However, to the best of our knowledge, no study has so far investigated and compared the efficacy of epoprostenol and several other cardiac drugs to treat acute heart failure in dogs.

This study aimed to evaluate the clinical utility of epoprostenol for treating PH in canine models of chronic PH. Additionally, we compared the effects of epoprostenol and various cardiac drugs for acute heart failure on RV hemodynamics and function. We hypothesized that epoprostenol might be an additional treatment option for PH in dogs.

## 2. Materials and Methods

This was a hypothesis-driven prospective observational study. All procedures followed the Guidelines for Institutional Laboratory Animal Care and Use of Nippon Veterinary and Life Science University in Japan, and the study was approved by the Ethics committee for Animal Use of Nippon Veterinary and Life Science University, Japan (approval number: 2022S-10).

### 2.1. Animals

Six male beagles owned by our laboratory (body weight: 11.0 ± 0.9 kg, age: 3.0 ± 0.3 years) were used in this study. Chronic PH models were developed by continuous injection of 150–300 µm diameter microsphere (Sephadex G-25 Coarse, Cytiva, Tokyo, Japan) into peripheral pulmonary arteries via multipurpose catheter surgically placed in the main pulmonary artery (median total microsphere injection: 1.56 g/kg [range: 1.22–1.71]) [14,16,17]. Chronic PH was defined as systolic PAP maintained above 50 mmHg for 4 weeks without the microsphere injection [16].

### 2.2. Study Protocol

All dogs were administered ampicillin sodium (20 mg/kg, IV, q2h; VMDP Co., Ltd., Tokyo, Japan). Anesthesia was induced with intravenous propofol injection (Nichi-Iko Pharmaceutical Co., Ltd., Toyama, Japan) and maintained with 1.5–2.0% isoflurane (Mylan Seiyaku Ltd., Osaka, Japan) mixed with 100% oxygen. Each dog received a continuous infusion of the drugs examined in this study or lactated Ringer’s solution at a rate of 1.0 mL/kg/h throughout the study protocol. Pressure-controlled mechanical ventilation was initiated at 8–10 breaths/minute. The end-tidal partial pressure of carbon dioxide, transcutaneously measured tissue oxygen saturation, heart rate and systemic arterial pressure (SAP) obtained using the oscillometric method were monitored throughout the study protocol with a multiparameter monitor (AM130, Fukuda M-E Kogyo Co., Ltd., Tokyo, Japan). All dogs were restrained in the left lateral recumbency. After clipping, aseptic preparation and draping around the right jugular vein, local anesthesia was induced with percutaneous injection of lidocaine around the right jugular vein. A 6-Fr sheath introducer (Radifocus Introducer IIH; Terumo Corporation, Tokyo, Japan) was inserted into the right jugular vein using the Seldinger retainment technique.

After a 15-min anesthetic stabilization period, right heart catheterization and transthoracic echocardiography were performed to establish baseline data. Subsequently, epoprostenol (Nichi-Iko Pharmaceutical Co., Ltd., Toyama, Japan) dissolved in a dedicated solution was infused at doses of 2, 5, 10, 15 and 20 ng/kg/min, referencing the recommended dose noted in the package insert and previous human reports [8,18]. The doses of epoprostenol were increased sequentially without a washout period. The same examinations as those at baseline were performed after the infusion of each dose of epoprostenol was continued for 10 min. After the measurements at the maximal dose of each study drug, a more than 15-min washout period was established to ensure that hemodynamic and echocardiographic variables had returned to the same level as at the beginning of the study protocol. Right heart catheterization and echocardiography were performed again to establish baseline data after the washout period. The dogs were then infused with 5 and 10 µg/kg/min dobutamine for 10 min and underwent the same examination as those performed at baseline [19]. Third, baseline measurements were taken after the washout period, and the same examinations as those performed at baseline were performed 10 min after 3 and 5 µg/kg/min dopamine infusion. Finally, after the washout period, right heart catheterization and echocardiography were performed before and 15 min after administering pimobendan (0.15 mg/kg, IV; Nippon Zenyaku Kogyo Co., Ltd., Koriyama, Fukushima, Japan).

When all study protocols were completed, the thermodilution catheter and sheath introducer were removed, and manual astriction was performed at the catheterization sites. All dogs were recovered from anesthesia. Subsequently, amoxicillin (20.0 mg/kg, PO, BID for 3 days; VMDP Co., Ltd., Tokyo, Japan) and robenacoxib (2.0 mg/kg, SC, SID as needed) were administered. After completing the study protocol, all the dogs were transferred to another study at our institution.

### 2.3. Right Heart Catheterization

Right heart catheterization was performed by a single investigator (YY) using a thermodilution catheter (Edwards Lifesciences Corporation, Tokyo, Japan). All hemodynamic data were randomly analyzed under the direction of another investigator using hemodynamic analysis software (LabChart Pro, version 7.3.8; ADInstruments, Nagoya, Aichi, Japan). Lead II electrocardiogram was simultaneously recorded. Using the thermodilution technique, a 3-mL ice-cold saline solution was rapidly injected from the proximal port three times to measure RV cardiac output and stroke volume (CO and SV, respectively).

All hemodynamic variables, except for RV CO and SV, were obtained from 10 consecutive sinus rhythms at the end-expiratory phase, and the mean values were used for statistical analysis. In this study, the following indices were measured as the RV hemodynamical variables: pulmonary artery wedge pressure (PAWP), PAP (systole, mean and diastole), RV pressure (RVP; systole and diastole) and the maximal and minimal first derivative of RVP (RV dP/dt_max_ and RV dP/dt_min_, respectively). Additionally, the right atrial pressure and central venous pressure were measured at the beginning and end of the study protocol. RV SV was obtained by dividing the RV CO by the heart rate calculated from the R-R interval at the time of RV CO measurements. RV CO and SV were normalized by body surface area, and the mean values from three measurements were used for statistical analysis.

### 2.4. Echocardiography

Two-dimensional and Doppler’s echocardiography were performed using an echocardiographic system (Vivid E95 Ultra Edition; GE Healthcare, Tokyo, Japan) and a 3.5–6.9 MHz transducer. Lead II electrocardiography was simultaneously performed, and data were displayed on the images. All data were obtained from at least five consecutive cardiac cycles in sinus rhythm during the end-expiratory phase. All echocardiographic data were randomly analyzed by the same investigator who performed right heart catheterization and by a well-trained cardiologist (YY) under the direction of another investigator using an offline workstation (EchoPAC PC, Version 204, GE Healthcare, Tokyo, Japan). This study used the mean values from three consecutive cardiac cycles for statistical analyses.

The following variables were obtained to evaluate the left ventricular (LV) morphology and function: LV volume, ejection fraction and LV SV and CO. The end-diastolic and end-systolic LV volume was measured using the biplane modified Simpson’s method and the left apical two- and four-chamber view and was normalized by body surface area (LVEDVI and LVESVI, respectively) [20,21]. LV SV was calculated using the cross-sectional area method as described previously [22]. LV CO was obtained by multiplying LV SV by the heart rate calculated from the R-R interval at the time of LV SV measurements.

In this study, end-diastolic and end-systolic RV areas (RVEDA and RVESA, respectively), RV fractional area change (FAC) and tricuspid annular plane systolic excursion (TAPSE) were measured using the left apical four-chamber view optimized for the right heart (RV focus view) [23,24]. The TAPSE was obtained using the B-mode method, as described previously [25,26]. The RV area, RV FAC and TAPSE were normalized by body weight (RVEDA index, RVESA index, RV FACn and TAPSEn, respectively), as described previously [26,27].
RVEDA index = (RVEDA [cm^2^])/(body weight [kg])^0.624^(1)
RVESA index = (RVESA [cm^2^])/(body weight [kg])^0.628^(2)
RV FACn = (RV FAC [%])/(body weight [kg])^−0.097^(3)
TAPSEn = (TAPSE [mm])/(body weight [kg])^0.284^(4)

Furthermore, in this study, we calculated systemic vascular resistance (SVR) and PVR using hemodynamic and echocardiographic variables.
SVR = (Mean SAP [mmHg])/(LV CO [L/min/m^2^])(5)
PVR = (Mean PAP [mmHg] − Mean PAWP [mmHg])/(RV CO [L/min/m^2^])(6)

The mean systemic arterial pressure was obtained using the oscillometric method, and the mean value of three measurements was used for statistical analyses [28].

In this study, two-dimensional speckle tracking echocardiography (2D-STE) was performed as a precise indicator of LV and RV function. As the LV functional variables, LV longitudinal and circumferential strain (LV-SL and LV SC, respectively) were measured using the left apical four-chamber view and the right parasternal short-axis view at the level of papillary muscle, respectively [29]. The RV longitudinal strain (RV-SL) was also measured as the RV functional variable using the RV focus view. The analytical procedures followed those described in our previous reports [14,15,25,30]. The RV-SL was obtained only from the RV free wall [14,25,30]. This study used the absolute value of the mean of the strain values obtained from three consecutive cardiac cycles for statistical analyses.

### 2.5. Statistical Analysis

All statistical analyses were performed using the EZR ver. 1.50 software (https://www.jichi.ac.jp/saitama-sct/SaitamaHP.files/statmed.html; accessed on 9 December 2022) [31]. Continuous data are reported as the mean ± standard deviation.

The normality of the data was evaluated using the Shapiro–Wilk test. The effects of each drug (epoprostenol, dobutamine and dopamine) on hemodynamic and echocardiographic variables were evaluated using a repeated-measures analysis of variance with subsequent pairwise comparisons using the Bonferroni-adjusted paired *t* test (normally distributed data) or Friedman rank sum test with subsequent pairwise comparisons using the Bonferroni-adjusted Wilcoxon signed-rank sum test (non-normally distributed data). Additionally, the effects of pimobendan on hemodynamic and echocardiographic variables were evaluated using a paired *t* test (normally distributed data) or Wilcoxon signed-rank sum test (non-normally distributed data). Furthermore, one-way analysis of variance with subsequent pairwise comparisons using the Bonferroni-adjusted Student’s *t* test (normally distributed data) or Kruskal-Wallis test with subsequent pairwise comparisons using the Bonferroni-adjusted Mann-Whitney U test (non-normally distributed data) was performed to compare systolic PAP and PVR after administration of the examined drug at the maximal dose. Statistical significance was set at *p* < 0.05 for all statistical analyses.

## 3. Results

All right heart catheterization- and echocardiography-derived variables were obtained at all time points. The right atrial pressure and central venous pressure at the beginning of the study protocol were 3.4 ± 0.9 mmHg and 3.2 ± 0.9 mmHg, respectively, and those at the end of the study protocol were 3.7 ± 0.9 mmHg and 3.2 ± 0.8 mmHg, respectively.

### 3.1. Epoprostenol Administration

The results of hemodynamic and echocardiographic variables are summarized in Table 1. PAWP, PAP, RVP and SAP showed no significant changes after epoprostenol administration. On the other hand, 15 to 20 ng/kg/min epoprostenol significantly decreased PVR and increased RV dP/dt_max_, RV CO and RV SV compared with baseline and 2–5 ng/kg/min epoprostenol. SVR decreased significantly after administering epoprostenol at 15 and 20 ng/kg/min.

No significant changes were observed in the LV and RV morphological indicators (LV volume and RV area) for echocardiographic variables. However, LV functional indicators, such as ejection fraction, LV SV, LV CO and 2D-STE-derived LV-SL and LV-SC, were significantly increased, especially after epoprostenol administration at 20 ng/kg/min. The RV-SL also increased with 15 and 20 ng/kg/min epoprostenol compared to baseline and 2 ng/kg/min epoprostenol (15 ng/kg/min: *p* = 0.048; 20 ng/kg/min: *p* = 0.048 and 0.04, respectively).

### 3.2. Dobutamine Administration

Table 2 shows the results of hemodynamic and echocardiographic variables after dobutamine administration. The systolic, mean and diastolic PAP significantly increased with dobutamine administration in a dose-dependent manner. Systolic RVP and RV dP/dt_max_ were also increased with 5 and 10 µg/kg/min dobutamine compared to baseline (systolic RVP: *p* = 0.03 and <0.01, respectively; RV dP/dt_max_: *p* = 0.049 and 0.045, respectively). Additionally, 5 µg/kg/min dobutamine increased PVR compared to baseline (*p* = 0.02), whereas 10 µg/kg/min dobutamine showed no significant difference from baseline (*p* = 0.98). RV CO significantly increased with 10 µg/kg/min dobutamine, and RV SV also increased with 5 and 10 µg/kg/min dobutamine administration. The 5 and 10 µg/kg/min dobutamine significantly increased systolic SAP compared to the baseline (*p* = 0.04 and 0.03, respectively). No significant difference was observed in SVR with dopamine administration.

For echocardiographic variables, LVESVI was significantly decreased, and ejection fraction, LV SL, and LV-SC increased significantly with dobutamine administration in a dose-dependent manner. LV SV increased significantly with 5 and 10 µg/kg/min dobutamine compared to baseline (both *p* < 0.01), and LV CO was significantly increased with 10 µg/kg/min dobutamine compared to baseline (*p* = 0.04). RV functional indicators, such as RV FACn, TAPSEn and RV-SL, also increased with dobutamine administration in a dose-dependent manner.

### 3.3. Dopamine Administration

Table 3 shows the results of hemodynamic and echocardiographic variables after dopamine administration. Systolic PAP and PVR increased significantly with 5 µg/kg/min dopamine compared to baseline (*p* = 0.02 and 0.01, respectively). Systolic and diastolic RVP and RV SV increased significantly with 3 and 5 µg/kg/min dopamine compared to baseline (systolic RVP: *p* = 0.048 and 0.02, respectively; diastolic RVP, both *p* = 0.02; RV SV, both *p* = 0.02). The RV dP/dt_max_ increased significantly with 3 and 5 µg/kg/min dopamine compared to baseline (both *p* = 0.048), and the RV dP/dt_min_ decreased significantly with 5 µg/kg/min dopamine compared to baseline (*p* = 0.049). The 5 µg/kg/min dopamine significantly increased systolic SAP and SVR compared with baseline (*p* = 0.02 and 0.01, respectively).

Although no significant change was observed in LV volume for echocardiographic variables, ejection fraction was significantly increased with 3 and 5 µg/kg/min dopamine compared to baseline (both *p* = 0.045). Additionally, 2D-STE-derived LV-SL and LV-SC increased significantly following dopamine administration in a dose-dependent manner. RV functional indicators, including RV FACn, TAPSEn and RV-SL, increased significantly with dopamine administration in a dose-dependent manner.

### 3.4. Pimobendan Administration

Table 4 shows the results of hemodynamic and echocardiographic variables after intravenous pimobendan administration. PAWP, PAP, diastolic RVP and PVR showed no significant changes with pimobendan administration. In contrast, systolic RVP, RV dP/dt_max_, RV CO and RV SV increased significantly, and RV dP/dt_min_ decreased significantly with pimobendan. SAP showed no significant change, while SVR decreased significantly with pimobendan.

For echocardiographic variables, significantly decreased LVESVI and RVESA indices were observed with pimobendan. Additionally, pimobendan significantly increased LV and RV functional indicators, except for TAPSEn.

### 3.5. Comparison of the Hemodynamical Effect of Each Cardiac Drug

Figure 1 shows the results of systolic PAP and PVR before and after administering each cardiac drug at the maximal dose. In this study, there were no significant differences in the systolic PAP and PVR at baseline before administering each drug (systolic PAP, *p* = 0.20; PVR, *p* = 0.64). In contrast, systolic PAP and PVR after 20 ng/kg/min epoprostenol administration were significantly lower than those after administration of 10 µg/kg/min dobutamine and 5 µg/kg/min dopamine (systolic PAP: *p* < 0.01, and 0.01, respectively; PVR: *p* = 0.03 and < 0.01, respectively). Additionally, a significant difference was observed in systolic PAP after 10 µg/kg/min dobutamine and pimobendan administration (*p* = 0.03).

## 4. Discussion

This study investigated the efficacy of epoprostenol, comparing it with various cardiac drugs for acute heart failure in a canine model of chronic PH. In this study, 15 to 20 ng/kg/min epoprostenol significantly decreased PVR and improved LV and RV function and pulmonary and systemic circulation without increasing PAP and RVP. These results suggested that epoprostenol might affect canine PH as well as human PH. Additionally, our pimobendan results supported the efficacy of pimobendan in canine PH, in line with a previous report. In contrast, although catecholamines (dobutamine and dopamine) significantly increased pulmonary and systemic circulation, the drugs also increased PAP and RVP. Our results indicate caution when using catecholamines in dogs with PH.

To the best of our knowledge, this is the first study to evaluate the effect of epoprostenol on left and right heart hemodynamics in canine models of chronic PH. Although 2 to 10 ng/kg epoprostenol showed no significant changes in hemodynamic and echocardiographic indices, higher doses (15 to 20 ng/kg/min) of epoprostenol showed significant positive effects, especially in PVR, SVR, LV and RV function, and cardiac output from the LV and RV without worsening of PAP through pulmonary and systemic vasodilating effects. These results support the clinical utility of epoprostenol in canine PH. However, the doses observed in this study were higher than those recommended for human PH [8]. The result was similar to our previous study of oral prostacyclin analog (beraprost sodium) in dogs, in which favorable pulmonary and systemic vasodilator effects were observed at a daily dose of 30 µg/kg (20.8 ng/kg/min in conversion) [14,15]. Therefore, our results indicate that species sensitivity to prostacyclin analogs may exist, and significant positive effects on canine PH are expected when used at higher doses than humans. However, since the present study only investigated the effects of epoprostenol up to 20 ng/kg/min, dose-dependent effects may be observed at further higher doses.

Regarding catecholamines, the inotropic effects of dobutamine and dopamine significantly increased LV and RV function and CO in a dose-dependent manner. These results indicate that catecholamine might be effective against cardiac dysfunction and circulatory insufficiency associated with PH. However, PAP and RVP also increased with catecholamine administration. These results suggest that catecholamines might not fundamentally improve PH pathophysiology and rather worsen RV loading conditions, which is in accordance with a previous report in human medicine [32]. Therefore, catecholamines might be effective for improving cardiac function and circulation in dogs with PH; however, careful monitoring should be recommended when using these drugs because of the possibility of worsening PH pathophysiology.

In this study, pimobendan administration significantly increased LV and RV cardiac function and CO without worsening PAP or epoprostenol levels. The results of PAP, PVR, LV and RV functional variables were consistent with those of a previous report of canine models of chronic embolic PH [5]. Although a significant decrease in PVR was not observed in this study, pimobendan increased LV and RV CO without increasing PAP or RVP, suggesting that it may be a safer inotropic agent than catecholamines for the treatment of PH. In contrast, this study showed that the pulmonary vasodilating effect of pimobendan was milder than that of epoprostenol. Therefore, epoprostenol is considered more suitable for the fundamental resolution of PH pathophysiology. In addition, the combination of epoprostenol and pimobendan may have a synergistic effect on PH treatment.

This study had several limitations. First, this study used canine models of chronic PH without symptoms of right heart failure (e.g., ascites, pleural effusion and pericardial effusion). The results might have changed in the clinical cases and those with more progressed PH. Second, this study could not monitor right atrial pressure and central venous pressure throughout the study protocol. Therefore, SVR cannot be measured accurately. Additionally, rapid infusion of iced saline solution was performed many times in this study, which might have caused differences in volume loading conditions when examining each drug. However, these pressures were not abnormally high and did not change between the beginning and the end of the study protocol. Therefore, the influence of the right atrial pressure and central venous pressure was considered minimal. Third, this study investigated and compared the effects of various drugs on hemodynamics and cardiac function in the same order. Although a washout period was established between each drug to ensure that hemodynamic and echocardiographic variables had returned to the same level as at the beginning of the study protocol, the influence of the drugs already administered might not be ruled out completely. Furthermore, because this study did not establish placebo control, the potential impact of this study protocol was unclear. Finally, we could not perform an a power calculation in this study, and the small sample size might have decreased the statistical power.

## 5. Conclusions

In conclusion, this study revealed that high-dose epoprostenol significantly decreased PVR and SVR and increased LV and RV cardiac function as well as CO in canine models of chronic PH. Although a significant decrease in PAP was not observed, epoprostenol could help effectively treat PH and associated right heart failure in dogs. In contrast, dobutamine and dopamine, empirically used to treat PH, significantly increased PAP, RVP, and LV and RV function. Although catecholamines would be effective in cardiac dysfunction attributable to PH, our results suggest that they might worsen PH pathophysiology, which encourages careful monitoring when using catecholamines for treating PH. Pimobendan, as well as epoprostenol, improved LV and RV function and CO without increasing PAP. However, improvements in PH pathophysiology through the vasodilating effect of pimobendan might be milder than that of epoprostenol. Further studies are needed to investigate the actual clinical cases of PH due to various causes and the efficacy of combination therapy with epoprostenol and inotropic drugs.

## Figures and Tables

**Figure 1 vetsci-10-00302-f001:**
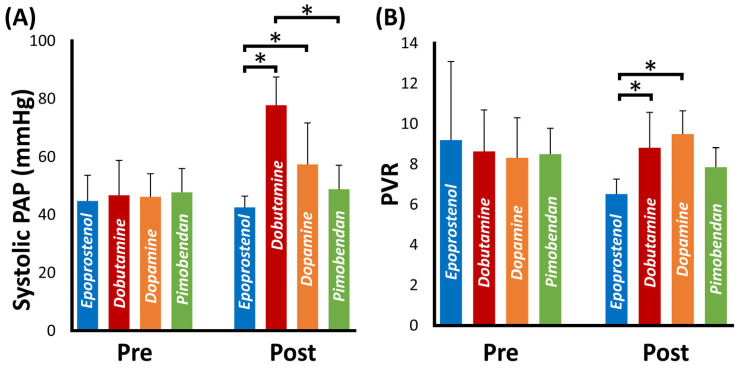
Results of systolic pulmonary arterial pressure (PAP) and pulmonary vascular resistance (PVR) before and after administration of various cardiac drugs at the maximal doses. Blue, red, orange and green bars represent the results of epoprostenol (20 ng/kg/min), dobutamine (10 µg/kg/min), dopamine (10 µg/kg/min) and pimobendan (0.15 mg/kg), respectively. * These values were significantly different (*p* < 0.05) based on the Bonferroni-adjusted Student’s *t* test (normally distributed data) or the Bonferroni-adjusted Mann-Whitney U test (non-normally distributed data).

**Table 1 vetsci-10-00302-t001:** Results of hemodynamic and echocardiographic variables before and after epoprostenol administration.

Variables	Baseline	Epoprostenol (ng/kg/min)					*p* *
		2	5	10	15	20	
Mean PAWP (mmHg)	7.2 ± 2.3	7.3 ± 1.9	6.9 ± 2.0	6.6 ± 2.0	6.6 ± 1.7	6.7 ± 1.9	0.72
Systolic PAP (mmHg)	44.8 ± 8.9	44.2 ± 10.1	45.9 ± 9.7	44.6 ± 9.6	43.2 ± 5.9	42.6 ± 3.9	0.14
Mean PAP (mmHg)	34 ± 8.8	33.9 ± 7.7	34.3 ± 7.8	35.6 ± 10.2	31.8 ± 5.5	31.5 ± 2.9	0.09
Diastolic PAP (mmHg)	24.8 ± 8.3	25.7 ± 6.7	25.2 ± 6.5	25.7 ± 7.7	22.7 ± 6.4	22.5 ± 4.2	0.12
Systolic RVP (mmHg)	46.4 ± 12.4	44.2 ± 12	43.5 ± 8.8	43.2 ± 9.4	42.8 ± 7	42.4 ± 7.3	0.02
Diastolic RVP (mmHg)	2.1 ± 1.4	1.7 ± 1.5	1.0 ± 1.9	0.8 ± 1.8	0.4 ± 1.9	1.2 ± 2.0	0.64
RV dP/dt_max_ (mmHg/s)	324 ± 52	340 ± 109	383 ± 94	384 ± 122	441 ± 103 ^a^	476 ± 119 ^a,b^	0.03
RV dP/dt_min_ (mmHg/s)	−374 ± 61	−359 ± 114	−437 ± 144	−424 ± 131	−505 ± 173	−540 ± 193	0.40
RV CO (L/min/m^2^)	2.7 ± 0.7	2.9 ± 0.9	3.2 ± 0.6	3.3 ± 0.6	3.7 ± 0.6 ^a^	3.8 ± 0.6 ^a,b,d^	<0.01
RV SV (mL/m^2^)	31.6 ± 6.9	33.5 ± 6.5	35.2 ± 5.8	33.0 ± 4.1	34.8 ± 5.7 ^a,b^	33.9 ± 5.1 ^a,b,c^	<0.01
Systolic SAP (mmHg)	108 ± 10	109 ± 13	100 ± 11	99 ± 10	100 ± 8	103 ± 12	0.64
Mean SAP (mmHg)	75 ± 10	73 ± 12	71 ± 9	69 ± 8	69 ± 7	72 ± 12	0.81
PVR	10.2 ± 3.9	9.7 ± 4.0	8.6 ± 2.4	8.7 ± 2.8	6.8 ± 1.4 ^a^	6.5 ± 0.7 ^a,b^	<0.01
SVR	17.5 ± 3.6	16.6 ± 3.5	14.8 ± 1.7	13.4 ± 0.8	13.0 ± 1.4 ^c^	12.8 ± 2.5 ^a,c^	<0.01
LVEDVI (mL/m^2^)	68.5 ± 10.7	70.3 ± 10.0	72.3 ± 12.9	71.8 ± 12.6	73.8 ± 10.2	71.4 ± 12.6	0.20
LVESVI (mL/m^2^)	37.6 ± 7.3	37.5 ± 6.2	39.7 ± 8.6	37.6 ± 9.3	36.3 ± 6.3	35.9 ± 7.4	0.39
Ejection fraction (%)	45.5 ± 3.5	46.6 ± 4.2	45.5 ± 2.6	48.2 ± 4.0	51.1 ± 2.5 ^a,c^	50.0 ± 3.5 ^a,c^	0.02
LV SV (mL/m^2^)	48.8 ± 7.3	48.2 ± 6.3	49.7 ± 6.6	50.6 ± 5.7	52.2 ± 4.2	53.8 ± 7.5 ^a,b,c^	<0.01
LV CO (L/min/m^2^)	4.4 ± 1	4.6 ± 1.1	4.9 ± 0.8	5.2 ± 0.8	5.4 ± 0.7	5.7 ± 0.6 ^a,b,c^	<0.01
RVEDA index (cm^2^/kg^0.624^)	1.5 ± 0.1	1.6 ± 0.2	1.6 ± 0.1	1.6 ± 0.1	1.6 ± 0.2	1.6 ± 0.1	0.13
RVESA index (cm^2^/kg^0.628^)	1.0 ± 0.1	1.1 ± 0.1	1.0 ± 0.1	1.0 ± 0.1	1.0 ± 0.1	1.0 ± 0.1	0.16
RV FACn (%/kg^−0.097^)	39.3 ± 4.4	40.1 ± 6.1	46.2 ± 3.8	47.3 ± 3.4	46.7 ± 2.9	49.4 ± 4.9	<0.01
TAPSEn (mm/kg^0.284^)	4.5 ± 0.6	4.9 ± 1.2	5.2 ± 1.0	5.6 ± 0.9	5.7 ± 1.1	5.8 ± 0.8 ^a,b^	<0.01
LV-SL (%)	12.1 ± 2.1	12.6 ± 2.1	12.7 ± 1.9	13.3 ± 2.2	13.6 ± 1.5 ^a^	14.5 ± 1.5 ^a,b,c^	<0.01
LV-SC (%)	14.0 ± 1.8	14.6 ± 1.8	14.8 ± 2.3	15.3 ± 2.2	16.1 ± 1.5 ^a^	16.6 ± 2.4 ^a,b,c^	<0.01
RV-SL (%)	17.5 ± 3.6	17.8 ± 3.2	19.3 ± 3.4	20.1 ± 3.2	21.2 ± 1.8 ^a,b^	22.5 ± 2.5 ^a,b^	<0.01

Data are represented as mean ± standard deviation. ^a^ Variables significantly differed from Baseline (*p* < 0.05). ^b^ Variables significantly differed from 2 ng/kg/min epoprostenol (*p* < 0.05). ^c^ Variables significantly differed from 5 µg/kg/min dobutamine (*p* < 0.05). ^d^ Variables significantly differed from 10 µg/kg/min dobutamine (*p* < 0.05). * *p*-value of repeated-measures analysis of variance (normally distributed data) or Friedman rank sum test (non-normally distributed data). Abbreviations: CO, cardiac output; dP/dt, first derivative of right ventricular pressure; FACn, RV fractional area change normalized by body weight; LV, left ventricular; LVEDVI, end-diastolic LV volume normalized by body surface area; LVESVI, end-systolic LV volume normalized by body surface area; LV-SC, LV circumferential strain; LV-SL, LV longitudinal strain; PAP, pulmonary arterial pressure; PAWP, pulmonary arterial wedge pressure; RV, right ventricular; RVEDA, end-diastolic RV area; RVESA, end-systolic RV area; RVP, RV pressure; RV-SL, RV longitudinal strain; SAP, systemic arterial pressure; SV, stroke volume; TAPSEn, tricuspid annular plane systolic excursion normalized by body weight.

**Table 2 vetsci-10-00302-t002:** Results of hemodynamic and echocardiographic variables before and after dobutamine administration.

Variables	Baseline	Dobutamine (µg/kg/min)		*p* *
		5	10	
Mean PAWP (mmHg)	7.4 ± 1.9	6.9 ± 2.0	7.0 ± 1.6	0.14
Systolic PAP (mmHg)	46.8 ± 12.0	65.2 ± 12.3 ^a^	77.8 ± 9.8 ^a,b^	<0.01
Mean PAP (mmHg)	35.2 ± 9.1	43.3 ± 11.6 ^a^	53.9 ± 7.6 ^a,b^	<0.01
Diastolic PAP (mmHg)	25.5 ± 7.3	28.8 ± 11.0	38.6 ± 6.9 ^a,b^	<0.01
Systolic RVP (mmHg)	46.8 ± 12.9	69.4 ± 19.1 ^a^	77.9 ± 16.4 ^a^	<0.01
Diastolic RVP (mmHg)	4.1 ± 4.6	5.2 ± 5.6	4.2 ± 7.6	0.74
RV dP/dt_max_ (mmHg/s)	434 ± 96	648 ± 254 ^a^	890 ± 390 ^a^	<0.01
RV dP/dt_min_ (mmHg/s)	−459 ± 112	−746 ± 289 ^a^	−923 ± 350 ^a,b^	<0.01
RV CO (L/min/m^2^)	3.2 ± 0.6	3.9 ± 1.2	5.5 ± 1.3 ^a,b^	<0.01
RV SV (mL/m^2^)	32.7 ± 3.1	44.0 ± 6.9 ^a^	46.2 ± 6.9 ^a^	<0.01
Systolic SAP (mmHg)	110 ± 14	129 ± 15 ^a^	127 ± 12 ^a^	<0.01
Mean SAP (mmHg)	76 ± 15	83 ± 16	85 ± 12	0.11
PVR	8.6 ± 2.1	9.5 ± 1.8 ^a^	8.8 ± 1.8	0.03
SVR	16.1 ± 3.3	17.2 ± 4.2	13.0 ± 2.4	0.09
LVEDVI (mL/m^2^)	72.5 ± 11.1	77.4 ± 12.2	73.0 ± 14.8	0.45
LVESVI (mL/m^2^)	41.0 ± 8.5	33.8 ± 7.0 ^a^	28.4 ± 7.1 ^a,b^	<0.01
Ejection fraction (%)	43.9 ± 3.6	56.6 ± 3.3 ^a^	61.4 ± 3.3 ^a,b^	<0.01
LV SV (mL/m^2^)	48.1 ± 7.9	61.3 ± 12.7 ^a^	64.2 ± 11.0 ^a^	<0.01
LV CO (L/min/m^2^)	4.8 ± 1.1	5.1 ± 1.6	6.4 ± 1.6 ^a^	0.02
RVEDA index (cm^2^/kg^0.624^)	1.5 ± 0.1	1.6 ± 0.1	1.5 ± 0.2	0.96
RVESA index (cm^2^/kg^0.628^)	1.0 ± 0.1	0.9 ± 0.2	0.8 ± 0.1 ^a^	0.03
RV FACn (%/kg^−0.097^)	44.3 ± 6.6	54.1 ± 7.3	58.0 ± 5.5 ^a^	0.02
TAPSEn (mm/kg^0.284^)	5.2 ± 1.0	6.5 ± 0.9 ^a^	6.8 ± 0.5 ^a^	<0.01
LV-SL (%)	12.8 ± 1.9	15.9 ± 1.9 ^a^	17.3 ± 1.5 ^a^	<0.01
LV-SC (%)	17.0 ± 2.6	25.2 ± 2.3 ^a^	29.4 ± 1.9 ^a,b^	<0.01
RV-SL (%)	15.2 ± 2.4	21.8 ± 2.2 ^a^	25.0 ± 1.3 ^a,b^	<0.01

Data are represented as mean ± standard deviation. ^a^ Variables significantly differed from Baseline (*p* < 0.05). ^b^ Variables significantly differed from 5 µg/kg/min dobutamine (*p* < 0.05). * *p*-value of repeated-measures analysis of variance (normally distributed data) or Friedman rank sum test (non-normally distributed data). Abbreviations: CO, cardiac output; dP/dt, first derivative of right ventricular pressure; FACn, RV fractional area change normalized by body weight; LV, left ventricular; LVEDVI, end-diastolic LV volume normalized by body surface area; LVESVI, end-systolic LV volume normalized by body surface area; LV-SC, LV circumferential strain; LV-SL, LV longitudinal strain; PAP, pulmonary arterial pressure; PAWP, pulmonary arterial wedge pressure; RV, right ventricular; RVEDA, end-diastolic RV area; RVESA, end-systolic RV area; RVP, RV pressure; RV-SL, RV longitudinal strain; SAP, systemic arterial pressure; SV, stroke volume; TAPSEn, tricuspid annular plane systolic excursion normalized by body weight.

**Table 3 vetsci-10-00302-t003:** Results of hemodynamic and echocardiographic variables before and after dopamine administration.

Variables	Baseline	Dopamine (µg/kg/min)		*p* *
		3	5	
Mean PAWP (mmHg)	7.7 ± 1.9	6.9 ± 2.3	6.4 ± 2.1	0.13
Systolic PAP (mmHg)	49.3 ± 7.9	51.4 ± 9.9	57.4 ± 14.3 ^a^	0.04
Mean PAP (mmHg)	36.8 ± 7.1	36.2 ± 6.5	39.7 ± 6.0	0.12
Diastolic PAP (mmHg)	29.3 ± 4.5	25.1 ± 4.7	25.0 ± 6.4	0.06
Systolic RVP (mmHg)	47.4 ± 8.1	51.7 ± 12.6 ^a^	55.4 ± 13.2 ^a^	0.03
Diastolic RVP (mmHg)	4.9 ± 4.4	3.5 ± 4.3 ^a^	2.3 ± 3.7 ^a^	<0.01
RV dP/dt_max_ (mmHg/s)	409 ± 127	511 ± 208 ^a^	499 ± 196 ^a^	0.04
RV dP/dt_min_ (mmHg/s)	−432 ± 126	−536 ± 204	−553 ± 203 ^a^	0.02
RV CO (L/min/m^2^)	3.5 ± 0.3	3.6 ± 0.7	3.5 ± 0.7	0.94
RV SV (mL/m^2^)	33.0 ± 4.1	41.1 ± 7.2 ^a^	46.8 ± 7.6 ^a^	<0.01
Systolic SAP (mmHg)	108 ± 10	110 ± 14	126 ± 10 ^a^	0.01
Mean SAP (mmHg)	75 ± 11	72 ± 11	75 ± 11	0.74
PVR	8.3 ± 2.0	8.3 ± 1.7	9.5 ± 1.2 ^a^	0.04
SVR	12.6 ± 0.9	14.4 ± 1.6	15.4 ± 1.4 ^a^	0.04
LVEDVI (mL/m^2^)	69.9 ± 11.6	70.7 ± 9.1	74.5 ± 8.5	0.12
LVESVI (mL/m^2^)	37.2 ± 8.5	34.4 ± 6.3	33.3 ± 7.6	0.11
Ejection fraction (%)	47.4 ± 4.3	51.7 ± 2.8 ^a^	55.7 ± 5.8 ^a^	<0.01
LV SV (mL/m^2^)	49.9 ± 4.9	55.0 ± 5.7	62.2 ± 5.5 ^a^	<0.01
LV CO (L/min/m^2^)	5.5 ± 0.5	5.0 ± 0.6	5.3 ± 1.1	0.38
RVEDA index (cm^2^/kg^0.624^)	1.6 ± 0.1	1.5 ± 0.2	1.6 ± 0.1	0.09
RVESA index (cm^2^/kg^0.628^)	1.1 ± 0.1	1.0 ± 0.1	0.9 ± 0.1 ^a^	0.04
RV FACn (%/kg^−0.097^)	39.8 ± 2.9	45.2 ± 2.8	53.7 ± 5.2 ^a,b^	<0.01
TAPSEn (mm/kg^0.284^)	4.9 ± 0.8	5.3 ± 0.6	6.4 ± 0.9 ^a,b^	<0.01
LV-SL (%)	12.6 ± 2.4	13.7 ± 1.4	16.2 ± 2.4 ^a,b^	<0.01
LV-SC (%)	14.9 ± 1.2	19.6 ± 3.2 ^a^	22.3 ± 2.0 ^a^	<0.01
RV-SL (%)	17.5 ± 2.0	21.5 ± 2.9 ^a^	25.1 ± 2.3 ^a,b^	<0.01

Data are represented as mean ± standard deviation. ^a^ Variables significantly differed from Baseline (*p* < 0.05). ^b^ Variables significantly differed from 3 µg/kg/min dopamine (*p* < 0.05). * *p*-value of repeated-measures analysis of variance (normally distributed data) or Friedman rank sum test (non-normally distributed data). Abbreviations: CO, cardiac output; dP/dt, first derivative of right ventricular pressure; FACn, RV fractional area change normalized by body weight; LV, left ventricular; LVEDVI, end-diastolic LV volume normalized by body surface area; LVESVI, end-systolic LV volume normalized by body surface area; LV-SC, LV circumferential strain; LV-SL, LV longitudinal strain; PAP, pulmonary arterial pressure; PAWP, pulmonary arterial wedge pressure; RV, right ventricular; RVEDA, end-diastolic RV area; RVESA, end-systolic RV area; RVP, RV pressure; RV-SL, RV longitudinal strain; SAP, systemic arterial pressure; SV, stroke volume; TAPSEn, tricuspid annular plane systolic excursion normalized by body weight.

**Table 4 vetsci-10-00302-t004:** Results of hemodynamic and echocardiographic variables before and after pimobendan administration.

Variables	Pimobendan (0.15 mg/kg)		*p* *
	Pre	Post	
Mean PAWP (mmHg)	7.4 ± 2.1	6.9 ± 1.8	0.35
Systolic PAP (mmHg)	47.8 ± 8.2	48.9 ± 8.2	0.53
Mean PAP (mmHg)	37.1 ± 5.5	37.5 ± 6.5	0.75
Diastolic PAP (mmHg)	28.2 ± 4.4	28.0 ± 5.3	0.86
Systolic RVP (mmHg)	44.7 ± 5.1	50.4 ± 10.3	0.04
Diastolic RVP (mmHg)	4.2 ± 4.3	1.8 ± 2.7	0.06
RV dP/dt_max_ (mmHg/s)	397 ± 124	480 ± 118	0.04
RV dP/dt_min_ (mmHg/s)	−418 ± 104	−512 ± 123	0.04
RV CO (L/min/m^2^)	3.5 ± 0.3	3.9 ± 0.4	0.03
RV SV (mL/m^2^)	34.4 ± 2.6	38.5 ± 3.5	0.01
Systolic SAP (mmHg)	109 ± 9	107 ± 8	0.82
Mean SAP (mmHg)	71 ± 3	70 ± 3	0.95
PVR	8.5 ± 1.3	7.8 ± 1.0	0.06
SVR	13.3 ± 2.2	11.6 ± 3.0	0.03
LVEDVI (mL/m^2^)	73.4 ± 9.4	73.0 ± 10.1	0.60
LVESVI (mL/m^2^)	38.8 ± 6.5	34.1 ± 6.7	<0.01
Ejection fraction (%)	47.4 ± 3.4	53.6 ± 3.2	<0.01
LV SV (mL/m^2^)	54.0 ± 6.9	61.3 ± 7.8	<0.01
LV CO (L/min/m^2^)	5.5 ± 0.9	6.4 ± 1.3	0.04
RVEDA index (cm^2^/kg^0.624^)	1.6 ± 0.2	1.5 ± 0.1	0.45
RVESA index (cm^2^/kg^0.628^)	1.1 ± 0.2	0.9 ± 0.1	0.04
RV FACn (%/kg^−0.097^)	42.0 ± 6.1	49.9 ± 5.3	<0.01
TAPSEn (mm/kg^0.284^)	5.4 ± 0.8	6.0 ± 0.5	0.07
LV-SL (%)	13.9 ± 1.8	15.5 ± 2.1	<0.01
LV-SC (%)	16.2 ± 1.9	19.5 ± 1.1	<0.01
RV-SL (%)	19.2 ± 2.9	23.0 ± 1.7	<0.01

Data are represented as mean ± standard deviation. * *p*-value of paired *t*-test (normally distributed data) or Wilcoxon signed-rank sum test (non-normally distributed data). Abbreviations: CO, cardiac output; dP/dt, first derivative of right ventricular pressure; FACn, RV fractional area change normalized by body weight; LV, left ventricular; LVEDVI, end-diastolic LV volume normalized by body surface area; LVESVI, end-systolic LV volume normalized by body surface area; LV-SC, LV circumferential strain; LV-SL, LV longitudinal strain; PAP, pulmonary arterial pressure; PAWP, pulmonary arterial wedge pressure; RV, right ventricular; RVEDA, end-diastolic RV area; RVESA, end-systolic RV area; RVP, RV pressure; RV-SL, RV longitudinal strain; SAP, systemic arterial pressure; SV, stroke volume; TAPSEn, tricuspid annular plane systolic excursion normalized by body weight.

## Data Availability

The datasets used or analyzed during the current study are available from the corresponding author upon reasonable request.

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
