# Peer review of "Cardiovascular Effect of Epoprostenol and Intravenous Cardiac Drugs for Acute Heart Failure on Canine Pulmonary Hypertension"

_vetsci, 2023, doi:10.3390/vetsci10040302_

Round 1

Reviewer 1 Report

Line 95 continuous 95 injection of 150–300 μm diameter microsphere into peripheral pulmonary arteries via 96 multipurpose catheter surgically placed in the main pulmonary artery  (Please specify the type, brand and manufacture of the microspheres used in the study. Dextran microsphere? the volume of injected microsphereas created an average pulmonary arterial pressure of how much?)

Line 116 echocardiography were performed to establish baseline data. Subsequently, epoprostenol (please insert transthoracic echocardiography)

Line 137 After completing the study protocol, all the dogs were  transferred to another study at our institution. (???????) in the research institution can the same dogs be included in other studies? What is the consequence of the induction of PAH by microspheres? Did the dogs remain hypertensive after the study? Please excuse me for my doubts but I'm not sure what to think about PAH induction

Author Response

Response to Reviewers

April 12th, 2023

Prof. Dr. Patrick Butaye

Editor-in-Chief

Veterinary Sciences

Dear Editors:

We wish to express our appreciation to the Editor and Reviewers for their insightful comments, which have helped us significantly improve our manuscript entitled “Cardiovascular Effect of Epoprostenol and Intravenous cardiac drugs for Acute Heart Failure on Canine Pulmonary Hypertension” (note that the title of our manuscript has been modified according to the Reviewer’s comment).

We attach here our revised manuscript, as well as a point-by-point response to the Reviewers’ comments. We hope that the revised paper meets your approval and will be further suitable for publication in the Veterinary Sciences as the Standard Article.

Our manuscript was revised according to the Reviewers’ comments, and the changes in the manuscript were shown by highlight text. Please see the main document and the following point-by-point responses.

Again, thank you for allowing us to further strengthen our manuscript with your valuable comments and queries. We have worked hard to incorporate your feedback and hope that these revisions persuade you to accept our submission.

Sincerely,

Ryohei Suzuki, DVM, PhD.

Laboratory of Veterinary Internal Medicine, School of Veterinary Medicine, Faculty of Veterinary Science, Nippon Veterinary and Life Science University, 1-7-1 Kyonan-cho, Musashino-shi, Tokyo, 180-8602, Japan.

Tel: +81-422-31-4151 (ext; 3435)

Fax: +81-422-31-5534

Response to Reviewers

Reviewer: 1

Comments to the Reviewer 1

We wish to express our strong appreciation to the reviewer for your insightful comments on our paper. We feel the comments have helped us significantly improve the paper. We hope that the revised paper meets your approval and will be more suitable for publication in the Veterinary Sciences as Standard Article. Please see the following point-by-point responses for details.

Comments to the Author

Line 95 “continuous injection of 150–300 μm diameter microsphere into peripheral pulmonary arteries via multipurpose catheter surgically placed in the main pulmonary artery” (Please specify the type, brand and manufacture of the microspheres used in the study. Dextran microsphere? the volume of injected microsphereas created an average pulmonary arterial pressure of how much?)

Response: Thank you very much for your comment. We have clarified the information about the microsphere used in this study. In this study, chronic PH was defined as systolic PAP maintained above 50 mmHg for 4 weeks without the microsphere injection. We have additionally included the sentence.

Line 96-101: Chronic PH models were developed for by continuous injection of 150–300 µm diameter microsphere (Sephadex G-25 Coarse, Cytiva, Tokyo, Japan) into peripheral pulmonary arteries via multipurpose catheter surgically placed in the main pulmonary artery (median total microsphere injection: 1.56 g/kg [range:1.22–1.71]). Chronic PH was defined as systolic PAP maintained above 50 mmHg for 4 weeks without the microsphere injection.

Line 116 “echocardiography were performed to establish baseline data. Subsequently, epoprostenol” (please insert transthoracic echocardiography)

Response: Thank you very much for your advice. We have added “transthoracic” according to the Reviewer’s comment.

Line 118-119: After a 15-minute anesthetic stabilization period, right heart catheterization and transthoracic echocardiography were performed to establish baseline data.

Line 137 “After completing the study protocol, all the dogs were transferred to another study at our institution.” (???????) in the research institution can the same dogs be included in other studies? What is the consequence of the induction of PAH by microspheres? Did the dogs remain hypertensive after the study? Please excuse me for my doubts but I'm not sure what to think about PAH induction

Response: Thank you very much for your comment. In our institution, we could use the same dogs for other studies. After completing the entire research protocol initially planned, the dogs will be euthanized, and histopathology will be performed. Almost all dogs remained high PAP. If PAP decreased, we injected the microspheres continuously again until the dog met the criteria of chronic PH.

Reviewer 2 Report

 The authors administered each of the following; epoprostenol (5 doses), dobutamine (2 doses), dopamine( 2 doses) and pimobendan (1 dose) in that order for 10 minutes, with a 15 minute washout between drugs to anesthetized ventilated beagles with artificially induced acute pulmonary hypertension (pulmonary artery microsphere injection). Right heart catheterization and echocardiography was used throughout to confirm washout and assess response to treatment.  These results do not support the use of catecholamines in treating pulmonary hypertension and suggest epoprostenol may be more effective than pimobendan.

There are benefits to the experimental  approach performed but it also has some limitations that were not addressed. The patients were anesthetized and ventilated for ~3 or more hours based on the experimental design. The order of agent administration was the same (not randomized) and no controls were performed to look at how the measured indices may have varied in the absence of pharmacologic manipulation. A washout brought patients back to a “baseline” between treatments, but may not reflect the true baseline without pharmacological manipulation. Experiments, data or evidence to negate the impact of drug order and treatment time should be added OR controls should be added that test these effects.

Similarly the investigators were not blinded on the treatment groups (or did not indicate that to be the case) there is a risk of bias with that experimental design that should be mentioned as an additional limitation. Alternatively, the data can be randomly reevaluated by a blinded investigator.

There is perhaps some excessive self-citation, can some of these be replaced by other evidence outside of the authors publications.

Some of the journal names are abbreviated while others have not been, this should be consistent and meet journal standards.

The order of agent administration (combined or not) is unclear in the abstract and should be reworded.

The title could be improved to indicate the focus was not on epoprostenol but a potential pulmonary hypertension therapeutics.

Overall, the paper is very well written and organized.

Author Response

Response to Reviewers

April 12th, 2023

Prof. Dr. Patrick Butaye

Editor-in-Chief

Veterinary Sciences

Dear Editors:

We wish to express our appreciation to the Editor and Reviewers for their insightful comments, which have helped us significantly improve our manuscript entitled “Cardiovascular Effect of Epoprostenol and Intravenous cardiac drugs for Acute Heart Failure on Canine Pulmonary Hypertension” (note that the title of our manuscript has been modified according to the Reviewer’s comment).We attach here our revised manuscript, as well as a point-by-point response to the Reviewers’ comments. We hope that the revised paper meets your approval and will be further suitable for publication in the Veterinary Sciences as the Standard Article.

Our manuscript was revised according to the Reviewers’ comments, and the changes in the manuscript were shown by highlight text. Please see the main document and the following point-by-point responses.

Again, thank you for allowing us to further strengthen our manuscript with your valuable comments and queries. We have worked hard to incorporate your feedback and hope that these revisions persuade you to accept our submission.

Sincerely,

Ryohei Suzuki, DVM, PhD.

Laboratory of Veterinary Internal Medicine, School of Veterinary Medicine, Faculty of Veterinary Science, Nippon Veterinary and Life Science University, 1-7-1 Kyonan-cho, Musashino-shi, Tokyo, 180-8602, Japan.

Tel: +81-422-31-4151 (ext; 3435)

Fax: +81-422-31-5534

Response to Reviewers

Reviewer: 2

Comments to the Reviewer 2

We wish to express our strong appreciation to the reviewer for your insightful comments on our paper. We feel the comments have helped us significantly improve the paper. We hope that the revised paper meets your approval and will be more suitable for publication in the Veterinary Sciences as Standard Article. Please see the following point-by-point responses for details.

The authors administered each of the following; epoprostenol (5 doses), dobutamine (2 doses), dopamine (2 doses) and pimobendan (1 dose) in that order for 10 minutes, with a 15 minute washout between drugs to anesthetized ventilated beagles with artificially induced acute pulmonary hypertension (pulmonary artery microsphere injection). Right heart catheterization and echocardiography was used throughout to confirm washout and assess response to treatment.  These results do not support the use of catecholamines in treating pulmonary hypertension and suggest epoprostenol may be more effective than pimobendan.

There are benefits to the experimental approach performed but it also has some limitations that were not addressed. The patients were anesthetized and ventilated for ~3 or more hours based on the experimental design. The order of agent administration was the same (not randomized) and no controls were performed to look at how the measured indices may have varied in the absence of pharmacologic manipulation. A washout brought patients back to a “baseline” between treatments, but may not reflect the true baseline without pharmacological manipulation. Experiments, data or evidence to negate the impact of drug order and treatment time should be added OR controls should be added that test these effects.

Response: Thank you very much for your comment. As you mentioned, this study investigated the effects of various drugs in the same order. Additionally, it was serious limitation that we did not set placebo control. We have added the limitation according to the Reviewer’s comment.

Line 413-419: Third, this study investigated and compared the effects of various drugs on hemodynamics and cardiac function in the same order. Although a washout period was established between each drug to ensure that hemodynamic and echocardiographic variables had returned to the same level as at the beginning of the study protocol, the influence of the drugs already administered might not be ruled out completely. Furthermore, because this study did not establish placebo control, the potential impact of this study protocol was unclear.

Similarly the investigators were not blinded on the treatment groups (or did not indicate that to be the case) there is a risk of bias with that experimental design that should be mentioned as an additional limitation. Alternatively, the data can be randomly reevaluated by a blinded investigator.

Response: Thank you very much for your comment. The investigator randomly analyzed hemodynamic and echocardiographic data under the direction of another investigator. We have added the information into the materials and methods.

Line 144-146: All hemodynamic data were randomly analyzed under the direction of another investigator using hemodynamic analysis software (LabChart Pro; ADInstruments, Aichi, Japan).

Line 166-169: All echocardiographic data were randomly analyzed by the same investigator who performed right heart catheterization and by a well-trained cardiologist (YY) under the direction of another investigator using an offline workstation (EchoPAC PC, Version 204, GE Healthcare, Tokyo, Japan).

There is perhaps some excessive self-citation, can some of these be replaced by other evidence outside of the authors publications.

Response: Thank you very much for your comment. We have avoided the self-citation and left only relevant references as possible. However, please note that there will inevitably be a lot of self-citation, especially in the 2D-STE methodology section.

Some of the journal names are abbreviated while others have not been, this should be consistent and meet journal standards.

Response: Thank you very much for your comment and we apologize for the inappropriate citation style. We have confirmed and modified the citation style. Please refer to the main document for the details.

The order of agent administration (combined or not) is unclear in the abstract and should be reworded.

Response: Thank you very much for your comment. We have clarified the content into the abstract. Additionally, we have changed some sentences due to the word limit. Please refer to the main document for the details.

Line 34-37: Six dogs with chronic PH were anesthetized and underwent right heart catheterization and echocardiography before and after infusion of epoprostenol, dobutamine, dopamine, and pimobendan (the drug administration order was the same for all dogs).

The title could be improved to indicate the focus was not on epoprostenol but a potential pulmonary hypertension therapeutics.

Response: Thank you very much for your comment. As you mentioned, this study aimed to investigate the cardiovascular effect of epoprostenol as well as that of catecholamines and pimobendan in canine models of chronic PH. We have modified the manuscript title according to the Reviewer’s comment.

Title: Cardiovascular Effect of Epoprostenol and Intravenous cardiac drugs for Acute Heart Failure on Canine Pulmonary Hypertension

Overall, the paper is very well written and organized.

Reviewer 3 Report

The article entitled Cardiovascular Effect of Intravenous Prostacyclin Analog (Epo-2 prostenol) on Canine Pulmonary Hypertension  is a well organized and clearly written work. I found the intricate topic of pulmonary pressures, pulmonary hypertension and the actions of the studied drugs to be properly and clearly explained with excellent results shown in table and graph formats. The tables and graphs are an great addition to the whole work and can be understood solely by reading their relative captions. I have only a few minor corrections :

In the second paragraph of the Introduction many sentences begin with the word however (line 63, 67, 69,and 71). Consider changing this using some synonyms such as: in addition, conversely, on the other hand.

Lines 74-76 - consider revising the phrase "… agent with a highly recommended level for patients…" to "…..agent highly recommended for patients …".

Line 169-171. You describe LV volume assessment using the modified Simpson's method, however the literature you cite is unrelated to this topic (13 and 14) or outdated (17). Please include relevant studies. Similarly in lines 174-177 the pertinent literature includes only positions 19 & 20.

line 355 consider revising sentence: This study investigated the efficacy of epoprostenol, comparing the various cardiac 354 drug for acute heart failure using canine models of chronic PH.

to: This study investigated the efficacy of epoprostenol, comparing it various cardiac drugs used for acute heart failure in a canine model of chronic PH.

Author Response

Response to Reviewers

April 12th, 2023

Prof. Dr. Patrick Butaye

Editor-in-Chief

Veterinary Sciences

Dear Editors:

We wish to express our appreciation to the Editor and Reviewers for their insightful comments, which have helped us significantly improve our manuscript entitled “Cardiovascular Effect of Epoprostenol and Intravenous cardiac drugs for Acute Heart Failure on Canine Pulmonary Hypertension” (note that the title of our manuscript has been modified according to the Reviewer’s comment).

We attach here our revised manuscript, as well as a point-by-point response to the Reviewers’ comments. We hope that the revised paper meets your approval and will be further suitable for publication in the Veterinary Sciences as the Standard Article.

Our manuscript was revised according to the Reviewers’ comments, and the changes in the manuscript were shown by highlight text. Please see the main document and the following point-by-point responses.

Again, thank you for allowing us to further strengthen our manuscript with your valuable comments and queries. We have worked hard to incorporate your feedback and hope that these revisions persuade you to accept our submission.

Sincerely,

Ryohei Suzuki, DVM, PhD.

Laboratory of Veterinary Internal Medicine, School of Veterinary Medicine, Faculty of Veterinary Science, Nippon Veterinary and Life Science University, 1-7-1 Kyonan-cho, Musashino-shi, Tokyo, 180-8602, Japan.

Tel: +81-422-31-4151 (ext; 3435)

Fax: +81-422-31-5534

Response to Reviewers

Reviewer: 3

Comments to the Reviewer 3

We wish to express our strong appreciation to the reviewer for your insightful comments on our paper. We feel the comments have helped us significantly improve the paper. We hope that the revised paper meets your approval and will be more suitable for publication in the Veterinary Sciences as Standard Article. Please see the following point-by-point responses for details.

The article entitled Cardiovascular Effect of Intravenous Prostacyclin Analog (Epoprostenol) on Canine Pulmonary Hypertension is a well organized and clearly written work. I found the intricate topic of pulmonary pressures, pulmonary hypertension and the actions of the studied drugs to be properly and clearly explained with excellent results shown in table and graph formats. The tables and graphs are an great addition to the whole work and can be understood solely by reading their relative captions. I have only a few minor corrections :

In the second paragraph of the Introduction many sentences begin with the word however (line 63, 67, 69,and 71). Consider changing this using some synonyms such as: in addition, conversely, on the other hand.

Response: Thank you very much for your advice. As you mentioned, this paragraph used "however" too many times. We have modified some sentence to avoid using the same word in many places.

Line 68-72: Conversely, since the effectiveness of catecholamines in treating PH has not been reported in veterinary medicine, they have been used empirically in the same way as in dogs with acute left-sided heart failure. Regarding intravenous pimobendan, a recent study has reported the effectiveness for the treatment of PH in dogs.

Lines 74-76 - consider revising the phrase "… agent with a highly recommended level for patients…" to "…..agent highly recommended for patients …".

Response: Thank you very much for your advice. We have revised the sentence according to the Reviewer’s comment.

Line 75-77: In human medicine, epoprostenol, an intravenous prostaglandin I2 (prostacyclin) analog, has been used to treat PH and is the only intravenous agent highly recommended for patients with severe PH [6,7].

Line 169-171. You describe LV volume assessment using the modified Simpson's method, however the literature you cite is unrelated to this topic (13 and 14) or outdated (17). Please include relevant studies. Similarly in lines 174-177 the pertinent literature includes only positions 19 & 20.

Response: Thank you very much for your comment. We have changed the appropriate references and left only relevant studies according to the Reviewer’s comment. If Reviewer considers that reference [17] should be deleted, we will delete it.

Line 173-176: The end-diastolic and end-systolic LV volume was measured using the biplane modified Simpson’s method and the left apical two- and four-chamber view and was normalized by body surface area (LVEDVI and LVESVI, respectively) [17,18].

  1. Schiller, N.B.; Shah, P.M.; Crawford, M.; DeMaria, A.; Devereux, R.; Feigenbaum, H.; Gutgesell, H.; Reichek, N.; Sahn, D.; Schnittger, I. Recommendations for Quantitation of the Left Ventricle by Two-Dimensional Echocardiography. American Society of Echocardiography Committee on Standards, Subcommittee on Quantitation of Two-Dimensional Echocardiograms. J Am Soc Echocardiogr 1989, 2, 358–367, doi:10.1016/s0894-7317(89)80014-8.
  2. Scollan, K.F.; Stieger-Vanegas, S.M.; David Sisson, D. Assessment of Left Ventricular Volume and Function in Healthy Dogs by Use of One-, Two-, and Three-Dimensional Echocardiography versus Multidetector Computed Tomography. Am J Vet Res 2016, 77, 1211–1219, doi:10.2460/AJVR.77.11.1211.

line 355 consider revising sentence: This study investigated the efficacy of epoprostenol, comparing the various cardiac drug for acute heart failure using canine models of chronic PH.

to: This study investigated the efficacy of epoprostenol, comparing it various cardiac drugs used for acute heart failure in a canine model of chronic PH.

Response: Thank you very much for your comment. We have modified the sentence according to the Reviewer’s comment.

Line 359-360: This study investigated the efficacy of epoprostenol, comparing it various cardiac drugs for acute heart failure in a canine model of chronic PH.

Round 2

Reviewer 2 Report

I thank the authors for their response and efforts to improve the quality of this already well written manuscript. The limitations to the study design have been acknowledged adequately. Despite these limitations I do consider this work to be of value and recommend publication. 

Author Response

Dear Reviewer 2

We express our strong appreciation to the Reviewer 2 for your insightful comments on our paper. We feel the comments have helped us significantly improve the paper. We are very honored that Reviewer 2 finally deemed our paper acceptable for publication in Veterinary Science as a standard article.